# Henneaux–Teitelboim Gauge Symmetry and Its Applications to Higher Gauge Theories



Mihailo Đorđević †, Tijana Radenković †, Pavle Stipsić † and Marko Vojinović *,†

Institute of Physics, University of Belgrade, Pregrevica 118, 11080 Belgrade, Serbia; mdjordjevic@ipb.ac.rs (M.Đ.); rtijana@ipb.ac.rs (T.R.); pstipsic@ipb.ac.rs (P.S.)
* Correspondence: vmarko@ipb.ac.rs
† These authors contributed equally to this work.

**Abstract:** When discussing the gauge symmetries of any theory, the Henneaux–Teitelboim transformations are often underappreciated or even completely ignored, due to their on-shell triviality. Nevertheless, these gauge transformations play an important role in understanding the structure of the full gauge symmetry group of any theory, especially regarding the subgroup of diffeomorphisms. We give a review of the Henneaux–Teitelboim transformations and the resulting gauge group in the general case and then discuss its role in the applications to the class of topological theories called *nBF* models, relevant for the constructions of higher gauge theories and quantum gravity.

**Keywords:** gauge symmetry; trivial gauge transformations; *nBF* theory; Chern–Simons theory; diffeomorphism symmetry





## 1. Introduction

In modern theoretical physics, gauge symmetries play a very prominent role. The two most-fundamental theories we have, which describe almost all observed phenomena in nature—namely Einstein's theory of general relativity and the Standard Model of elementary particle physics—are gauge theories. From Maxwell's electrodynamics to various approaches to quantum gravity, gauge theories play a central role, and gauge symmetry represents one of their most-important aspects. In light of this, there is one class of gauge transformations that is often slightly neglected in the literature, due to their specific nature and properties.

In order to introduce this particular gauge symmetry in the most-elementary way possible, let us look at the following simple example. Every action $S[\phi_1, \phi_2]$, which depends on the fields $\phi_1(x)$ and $\phi_2(x)$, is invariant under the following gauge transformation:

$$\delta_0 \phi_1(x) = \epsilon(x) \frac{\delta S}{\delta \phi_2(x)}, \qquad \delta_0 \phi_2(x) = -\epsilon(x) \frac{\delta S}{\delta \phi_1(x)}, \tag{1}$$

as one can see by calculating the variation of the action:

$$\delta S[\phi_1, \phi_2] = \frac{\delta S}{\delta \phi_1} \delta_0 \phi_1 + \frac{\delta S}{\delta \phi_2} \delta_0 \phi_2 = 0. \tag{2}$$

This gauge symmetry exists for every action that is a functional of at least two fields, irrespective of any other gauge symmetry that the action may or may not have. In the literature, this symmetry is often called *trivial* gauge symmetry, since the form variations of the fields are identically zero on-shell. This is in contrast to all other gauge symmetries, which perform some nontrivial change of the fields on-shell.

It should be noted that, being trivial on-shell, the above transformations cannot play a role in obtaining any predictions about observables in a given theory, due to the intrinsic on-shell nature of the physical observables. For example, in practical situations

of scattering experiments and measurements of cross-sections, this trivial symmetry is irrelevant. Nevertheless, when constructing a new theory, in general, the off-shell properties of the theory are important. As a typical example, path integral quantization prescription depends not only on the classical equations of motion, but on the whole action of the theory. In this sense, while these trivial transformations are not relevant for making predictions, they do have methodological relevance and value in theory construction, despite their on-shell triviality.

For example, these transformations in fact represent a very important part of the gauge symmetry for any theory and play a crucial role in various contexts, such as in the Batalin–Vilkovisky formalism (see [1] for a review and also the original papers [2–6]), or when discussing the diffeomorphism symmetry of the *BF*-like class of theories [7–11]. Furthermore, in general, a commutator of two ordinary gauge transformations will remain an ordinary gauge transformation only up to the above trivial transformations, meaning that the latter are important for the algebraic closure of all gauge transformations into a group.

To the best of our knowledge, the most-complete treatment and discussion of the above gauge transformations can be found in the book [12] by Marc Henneaux and Claudio Teitelboim. Therefore, in this paper, we opted to call them Henneaux–Teitelboim (HT) transformations. This naming can also be justified with the paper [7] by Gary Horowitz (published two years before the book [12]), where the author attributes these transformations to Henneaux and Teitelboim in a footnote and thanks them "for explaining this to me".

Regarding terminology, we should also note that we use the terms "gauge symmetry" and "gauge transformations" with a certain level of charity. Namely, one could argue that there are two distinct types of local symmetries—those that are obtained by a localization procedure from a corresponding global symmetry group (the procedure of "gauging" a global symmetry) and those that are intrinsically local, not obtained by any such localization procedure. It is not known whether HT symmetry belongs to the former or the latter class, since a global symmetry whose localization would give rise to HT transformations has not yet been shown to exist. Either way, in the literature, there is no established terminology that distinguishes the two classes of symmetries, and most often, both are called "gauge symmetries". Therefore, in what follows, for a lack of better terminology, we will adhere to this practice and describe HT transformations as a gauge symmetry.

In some of the modern approaches to the problem of quantum gravity based on the spinfoam formalism of loop quantum gravity [13,14], as well as in other applications of the so-called higher gauge theory (see [15] for a review and [16] for an application to quantum gravity), the description of gauge symmetry is being extended from the notion of a Lie group to different algebraic structures, called 2-groups, 3-groups, and in general, $n$-groups [17–27]. In this context, it is important to revisit and study the specific class of HT gauge symmetries, since they provide a nontrivial insight into the properties of these more general algebraic structures, as well as the physics behind the symmetries they describe.

The purpose of this paper is to provide a review of HT transformations in general and then discuss their properties and applications in two concrete models—the Chern–Simons theory and the $3BF$ theory. The Chern–Simons case is simple enough to serve as an illustrative toy example, while the $3BF$ theory represents a basis for the construction of a realistic theory of quantum gravity with matter within the context of the spinfoam formalism (see also [16,28–32]), discussing that its HT symmetry represents an important stepping stone towards the goal of a more realistic theory. The main result of this work represents a clarification of the structure of the gauge symmetry of a pure topological $3BF$ action, as well as the corresponding symmetry for the constrained $2BF$ action, which is classically equivalent to Einstein's general relativity. We also discuss in detail the relationship between diffeomorphism symmetry and the HT symmetry for the Chern–Simons and $3BF$ theories and offer some conceptual suggestions regarding the notion of gauge symmetry as it is being used in the literature.

The layout of the paper is as follows. In Section 2, we give a review of the general theory of HT transformations and their main properties. Section 3 is devoted to the example of HT symmetry in Chern–Simons theory, which is convenient due to its simplicity. In Section 4, we discuss the main case of HT symmetry in the $3BF$ and $2BF$ theories, which are important for applications in quantum gravity models. Finally, Section 5 contains an overview of the results, future research directions, and some concluding remarks.

The notation and conventions in the paper are as follows. When important, we assume the $(-, +, +, +)$ signature of the spacetime metric. The Greek indices from the middle of the alphabet, $\lambda, \mu, \nu, \ldots$, represent spacetime indices and take values $0, 1, \ldots, D - 1$, where $D$ is the dimension of the spacetime manifold $\mathcal{M}_D$ under consideration. The Greek indices from the beginning of the alphabet, $\alpha, \beta, \gamma, \ldots$, represent group indices, as well as Latin indices $a, b, c, \ldots$ and uppercase Latin indices $A, B, C, \ldots$ and $I, J, K, \ldots$. All these indices will be assigned to various Lie groups under consideration. Lowercase Latin indices from the middle of the alphabet, $i, j, k, \ldots$, are generic and will be used to count all fields in a given theory or for some other purpose depending on the context. Throughout the paper, we denote the space of algebra-valued differential $p$-forms as

$$\mathcal{A}^p(\mathcal{M}, \mathfrak{a}) \equiv \Lambda^p(\mathcal{M}) \otimes \mathfrak{a},$$

where $\Lambda^p(\mathcal{M})$ is the ordinary space of differential $p$-forms over the manifold $\mathcal{M}$, while $\mathfrak{a}$ is some Lie algebra.

## 2. Review of HT Symmetry

We begin by studying some basic general properties of HT transformations. After the definition, we demonstrate that the group of HT transformations represents a normal subgroup of the *total* gauge group of a given theory, and we discuss the triviality of HT transformations and that they exhaust all possible trivial transformations. Finally, before moving on to concrete theories, we study the subtleties of the dependence of HT symmetry on the choice of the action.

### 2.1. Definition of HT Transformations

Given an action $S[\phi^i]$ as a functional of fields $\phi^i(x)$ ($i \in \{1, \ldots, N\}$ where we assume $N \geqslant 2$), the infinitesimal HT transformation is defined as

$$\phi^i(x) \to \phi'^i(x) = \phi^i(x) + \delta_0 \phi^i(x), \tag{3}$$

where the form variations of the fields are defined as

$$\delta_0 \phi^i(x) = \epsilon^{ij}(x) \frac{\delta S}{\delta \phi^j(x)}. \tag{4}$$

The variation of the action under HT transformations then gives

$$\delta S = \frac{\delta S}{\delta \phi^i} \delta_0 \phi^i = \frac{\delta S}{\delta \phi^i} \frac{\delta S}{\delta \phi^j} \epsilon^{ij}. \tag{5}$$

If the HT parameters are chosen to be antisymmetric,

$$\epsilon^{ij}(x) = -\epsilon^{ji}(x), \tag{6}$$

the variation of the action (5) is identically zero, and HT transformations (4) represent a gauge symmetry of the theory.

The most-striking thing in the above definition is the fact that we did not specify the action in any way. Aside from the assumption $N \geqslant 2$, which excludes only actions describing a single real scalar field, every action is invariant with respect to the HT transformations. In other words, *HT transformations are a gauge symmetry of essentially every theory.*

The second striking property of the definition is that the form variations of fields become zero on-shell, according to (4). In this sense, the HT symmetry is sometimes called *trivial symmetry*, in contrast to ordinary gauge symmetries that a theory may have, which transform the fields in a nontrivial way on-shell. Triviality is also the reason why HT gauge symmetry does not feature in any way in the Hamiltonian analysis of a theory, so only the presence of ordinary gauge symmetries can be deduced from the Hamiltonian formalism.

### 2.2. HT Symmetry Group and Its Properties

There are two general properties that can be formulated for HT transformations. The first is that HT transformations form a normal subgroup within the full group of gauge symmetries, while the second is that HT transformations exhaust the set of all possible trivial transformations. The consequence of these properties is that one can always write the total symmetry group of any theory as

$$\mathcal{G}_{\text{total}} = \mathcal{G}_{\text{nontrivial}} \ltimes \mathcal{G}_{HT} \,, \tag{7}$$

where $\mathcal{G}_{\text{nontrivial}}$ is the symmetry group of ordinary gauge transformations (if there are any), $\mathcal{G}_{HT}$ is the HT symmetry group, and the symbol $\ltimes$ stands for a semidirect product. One can also reformulate (7) as

$$\mathcal{G}_{\text{nontrivial}} = \mathcal{G}_{\text{total}} / \mathcal{G}_{HT} \,, \tag{8}$$

so that the group of ordinary gauge symmetries is represented as a quotient group.

The easiest way to demonstrate (7) is to prove that the Lie algebra corresponding to $\mathcal{G}_{HT}$ represents an ideal within the Lie algebra corresponding to $\mathcal{G}_{\text{total}}$. To that end, pick an arbitrary form variation of fields that represents a symmetry of the action and write it in the form

$$\hat{\delta}_0 \phi^i(x) = F^i(x) \,, \qquad \text{such that} \qquad \hat{\delta} S = \frac{\delta S}{\delta \phi^i} F^i \equiv 0 \,. \tag{9}$$

Then, using (4), we can take concatenated variations of this form variation and the HT form variation as

$$\delta_0 \hat{\delta}_0 \phi^i = \frac{\delta F^i}{\delta \phi^j} \frac{\delta S}{\delta \phi^k} \epsilon^{jk} \,,$$

and

$$\hat{\delta}_0 \delta_o \phi^i = \frac{\delta}{\delta \phi^k} \left( \epsilon^{ij} \frac{\delta S}{\delta \phi^j} \right) F^k = \frac{\delta \epsilon^{ij}}{\delta \phi^k} \frac{\delta S}{\delta \phi^j} F^k + \epsilon^{ij} \frac{\delta}{\delta \phi^j} \left( \frac{\delta S}{\delta \phi^k} F^k \right) - \epsilon^{ij} \frac{\delta S}{\delta \phi^k} \frac{\delta F^k}{\delta \phi^j} \,.$$

The term in the second parentheses is zero by (9), so the commutator of two-form variations becomes

$$[\delta_0, \hat{\delta}_0] \phi^i = \left( \epsilon^{jk} \frac{\delta F^i}{\delta \phi^j} - \epsilon^{ji} \frac{\delta F^k}{\delta \phi^j} - \frac{\delta \epsilon^{ik}}{\delta \phi^j} F^j \right) \frac{\delta S}{\delta \phi^k} \,, \tag{10}$$

which is again an HT transformation, since the expression in the parentheses is antisymmetric with respect to indices $i, k$. Therefore, the commutator is always an element of HT algebra, which means that HT algebra itself is an ideal of the total symmetry algebra. At the Lie group level, this translates into (7).

The second general property is the statement that there are no other trivial transformations beside the HT transformations. Assuming that some transformation described by the form variation $\bar{\delta}_0 \phi^i$ is a gauge symmetry of the action that vanishes on-shell, i.e., that it satisfies

$$\frac{\delta S}{\delta \phi^i} \bar{\delta}_0 \phi^i = 0 \,, \qquad \text{and} \qquad \bar{\delta}_0 \phi^i \approx 0 \,,$$

then one can prove that this transformation is an HT transformation, i.e., there exists a choice of antisymmetric HT parameters $\epsilon^{ij}$ such that the form variation $\bar{\delta}_0 \phi^i$ is of type (4):

$$\bar{\delta}_0 \phi^i = \epsilon^{ij} \frac{\delta S}{\delta \phi^j} \,. \tag{11}$$

Provided certain suitable regularity conditions for the action $S$, this statement can be rigorously formulated as a theorem. However, we omitted the proof since it is technical and off topic for the purposes of this paper. The interested reader can find the details of both the theorem and the proof in [12], Appendix 10.A.2.

To sum up, the first property (10) tells us that one can always factorize the total gauge symmetry group into the form (7), while the second property (11) guarantees that the quotient group (8) contains only nontrivial gauge transformations. This factorization of the total symmetry group is a key result that lays the groundwork for any subsequent analysis of HT transformations in particular and gauge symmetry in general.

*2.3. Dependence of HT Symmetry on the Action*

The final property of HT transformations that needs to be discussed is their dependence on the choice of the action. Suppose we are given some action $S_{\text{old}}[\phi^i]$, where $i \in \{1, \ldots, N\}$, which has the corresponding HT transformation described as in (4):

$$\delta_0^{\text{old}} \phi^i = \epsilon^{ij} \frac{\delta S_{\text{old}}}{\delta \phi^j} \, . \tag{12}$$

Now, suppose that we modify that action into another one, $S_{\text{new}}[\phi^i, \chi^k]$, where $k \in \{N + 1, \ldots, N + M\}$, by adding an extra term to the old action:

$$S_{\text{new}}[\phi^i, \chi^k] = S_{\text{old}}[\phi^i] + S_{\text{extra}}[\phi^i, \chi^k] \, . \tag{13}$$

Here, $\chi^j$ are additional fields that may be introduced into the new action. The HT transformation corresponding to the new action can be written in the block-matrix form, made of blocks of sizes $N$ and $M$, as follows:

$$\begin{pmatrix} \delta_0^{\text{new}} \phi^i \\ \delta_0^{\text{new}} \chi^k \end{pmatrix} = \left( \begin{array}{c|c} \epsilon^{ij} & \zeta^{il} \\ \hline \theta^{kj} & \psi^{kl} \end{array} \right) \begin{pmatrix} \frac{\delta S_{\text{new}}}{\delta \phi^j} \\ \frac{\delta S_{\text{new}}}{\delta \chi^l} \end{pmatrix} , \qquad \begin{array}{l} i,j \in \{1, \ldots, N\} \, , \\ k,l \in \{N + 1, \ldots, N + M\} \, . \end{array} \tag{14}$$

Here, $\epsilon = -\epsilon^T$ is an antisymmetric $N \times N$ block of parameters $\epsilon^{ij}$, $\zeta$ is a rectangular $N \times M$ block of parameters $\zeta^{il}$, $\theta$ is a rectangular $M \times N$ block such that $\theta = -\zeta^T$, and finally, $\psi = -\psi^T$ is an antisymmetric $M \times M$ block of parameters $\psi^{kl}$. Overall, the total parameter matrix is antisymmetric, as required by (6).

The question one can now study is what is the relation between the two HT gauge symmetry groups $\mathcal{G}_{HT}^{\text{old}}$ and $\mathcal{G}_{HT}^{\text{new}}$ that correspond to the two actions. In practice, this question is most often relevant in cases when one introduces the piece $S_{\text{extra}}$ as a gauge-fixing term, whose purpose is to break the ordinary gauge symmetry down to its subgroup:

$$G_{\text{nontrivial}}^{\text{new}} \subset G_{\text{nontrivial}}^{\text{old}} \, .$$

Naively, one might expect a similar relationship between the HT symmetry groups, $\mathcal{G}_{HT}^{\text{new}} \subset \mathcal{G}_{HT}^{\text{old}}$. However, looking at (12) and (14), this is obviously wrong. Namely, if $M \geqslant 1$, the HT symmetry of the new action is *larger* than the HT symmetry of the old action. Counting the number of independent parameters of both, one easily sees that

$$\dim(\mathcal{G}_{HT}^{\text{old}}) = \frac{N(N-1)}{2} \, , \qquad \dim(\mathcal{G}_{HT}^{\text{new}}) = \frac{(N+M)(N+M-1)}{2} \, ,$$

so that the only possible relationship would be the opposite, $\mathcal{G}_{HT}^{\text{old}} \subset \mathcal{G}_{HT}^{\text{new}}$. However, in fact, this can also be shown to be wrong. Namely, one can choose the extra parameters $\zeta$, $\theta$ and $\psi$ to be zero in (14), reducing it to the form that is formally similar to (12):

$$\delta_0^{\text{new}} \phi^i = \epsilon^{ij} \frac{\delta S_{\text{new}}}{\delta \phi^j} \, .$$

However, taking into account the relationship (13) between the two actions, the HT transformation takes the form

$$\delta_0^{\text{new}}\phi^i = \epsilon^{ij}\frac{\delta S_{\text{old}}}{\delta\phi^j} + \epsilon^{ij}\frac{\delta S_{\text{extra}}}{\delta\phi^j},$$

which is explicitly different from (12), due to the presence of the term $S_{\text{extra}}$ in the action. Therefore, the gauge group $\mathcal{G}_{HT}^{\text{old}}$ is not a subgroup of $\mathcal{G}_{HT}^{\text{new}}$ either.

The overall conclusion is that introducing additional terms to the action changes the total gauge symmetry in a nontrivial way. On the one hand, the ordinary gauge symmetry group typically becomes *smaller* due to explicit symmetry breaking by the extra term. On the other hand, the HT gauge symmetry group may become *larger* if the extra term contains additional fields, but either way becomes *different*, as a consequence of the very presence of the extra term. Given this, one can conclude that the *total* symmetry groups for the two actions will always be mutually different:

$$\mathcal{G}_{\text{total}}^{\text{new}} = \mathcal{G}_{\text{nontrivial}}^{\text{new}} \ltimes \mathcal{G}_{HT}^{\text{new}} \qquad \neq \qquad \mathcal{G}_{\text{total}}^{\text{old}} = \mathcal{G}_{\text{nontrivial}}^{\text{old}} \ltimes \mathcal{G}_{HT}^{\text{old}}.$$

Specifically, one cannot claim that the group $\mathcal{G}_{\text{total}}^{\text{old}}$ is being broken down into $\mathcal{G}_{\text{total}}^{\text{new}}$ as its subgroup; such a relationship may hold exclusively for the quotient groups of ordinary gauge transformations.

In the next two sections, we will turn to explicit examples of all general properties and features of the HT symmetry that have been discussed above. Moreover, we will also discuss some additional particular properties, such as the fact that some nontrivial gauge subgroups of $\mathcal{G}_{\text{total}}$ are not simultaneously subgroups of $\mathcal{G}_{\text{nontrivial}}$, which is a consequence of the semidirect product in (7). One such example will be the diffeomorphism symmetry in the Chern–Simons and $3BF$ actions.

Let us conclude this section with one conceptual comment. Throughout the literature, the typical practice is to always take the quotient between the total and HT symmetry groups as in (8), in order to isolate the nontrivial gauge transformations, and call the latter simply as the "gauge symmetry" of a theory. This approach is in fact advocated for in [12]. However, we believe that this practice can be misleading and that one should instead describe the group $\mathcal{G}_{\text{total}}$ as "the gauge symmetry" of a theory, explicitly including the HT subgroup as a legitimate gauge symmetry group. Namely, despite the fact that it is often called "trivial", the consequences of its presence in $\mathcal{G}_{\text{total}}$ are far from trivial. Granted, it may often be enough to discuss the gauge symmetry on-shell, and then, one can indeed calculate all symmetry transformations only "up to equations of motion", with no mention of the HT subgroup. However, whenever one needs to discuss the gauge transformations off-shell, the HT subgroup simply cannot be ignored anymore. Typical situations include the Batalin–Vilkovisky formalism [1], various generalizations of gauge symmetry in the context of higher gauge theories and quantum gravity [33], and even the traditional contexts such as the Coleman–Mandula theorem [34]. The situations in which HT transformations play a significant role may be rare, but nevertheless, they tend to be important. Thus, in our opinion, it would be prudent to always be aware that, for any given theory, its total gauge symmetry group is in fact bigger, and more feature-rich, than just the group of ordinary gauge transformations that are typically discussed in the literature.

## 3. HT Symmetry in Chern–Simons Theory

As an illustrative example of the general properties of HT symmetry from the previous section, let us discuss the HT transformations for the simple case of the Chern–Simons theory. The Chern–Simons theory represents an excellent toy example since it is well known in the literature and most readers should be familiar with it.

Given any Lie group $G$, its corresponding Lie algebra $\mathfrak{g}$, and a three-dimensional manifold $\mathcal{M}_3$, the Chern–Simons theory can be defined as a topological field theory over a trivial principal bundle $G \to \mathcal{M}_3$, given by the action:

$$S_{CS} = \int_{\mathcal{M}_3} \langle A \wedge \mathrm{d}A \rangle_{\mathfrak{g}} + \frac{1}{3} \langle A \wedge [A \wedge A] \rangle_{\mathfrak{g}}. \tag{15}$$

Here, $A \in \mathcal{A}^1(\mathcal{M}_3, \mathfrak{g})$ is a $\mathfrak{g}$-valued connection one-form over a manifold $\mathcal{M}_3$, and $\langle \_, \_ \rangle_{\mathfrak{g}}$ is a $G$-invariant symmetric nondegenerate bilinear form on $\mathfrak{g}$. One often rewrites the Chern–Simons action within the framework of the enveloping algebra of $\mathfrak{g}$, introducing the notion of a *trace* as

$$\mathrm{Tr}(XY) \equiv \langle X, Y \rangle_{\mathfrak{g}},$$

for every $X, Y \in \mathfrak{g}$. Then, the Chern–Simons action can be rewritten as

$$S_{CS} = \int_{\mathcal{M}_3} \mathrm{Tr}\left( A \wedge \mathrm{d}A + \frac{2}{3} A \wedge A \wedge A \right), \tag{16}$$

where, for the second term, one employs the identity $\mathrm{Tr}(X[Y, Z]) = \mathrm{Tr}(XYZ) - \mathrm{Tr}(XZY)$ for every $X, Y, Z \in \mathfrak{g}$.

The gauge symmetry of the Chern–Simons action consists of $G$-gauge transformations, determined with the parameters $\epsilon_{\mathfrak{g}}{}^I(x)$. Using the basis of generators $T_I$ to expand the connection $A$ into components as

$$A = A^I{}_\mu(x)\, \mathrm{d}x^\mu \otimes T_I,$$

the form variation of the connection components $A^I{}_\mu$ corresponding to gauge transformations can then be written as

$$\delta_0 A^I{}_\mu = \partial_\mu \epsilon_{\mathfrak{g}}{}^I - f_{JK}{}^I \epsilon_{\mathfrak{g}}{}^J A^K{}_\mu, \tag{17}$$

where $f_{JK}{}^I$ are the structure constants corresponding to the generators $T_I$. Therefore, the gauge symmetry of the Chern–Simons theory is usually quoted as the initially chosen Lie group $G$:

$$\mathcal{G}_{CS} = G. \tag{18}$$

However, as we have seen in the previous section, this is not the complete set of gauge transformations, and the *total* gauge group should in fact be

$$\mathcal{G}_{\text{total}} = \mathcal{G}_{CS} \ltimes \mathcal{G}_{HT}. \tag{19}$$

Let us define the HT transformations for the Chern–Simons action (15). If we denote the dimension of the Lie algebra $\mathfrak{g}$ as $\dim(\mathfrak{g}) = p$, the number of independent field components $A^I{}_\mu$ is $N = 3p$. The HT transformation is then defined with the HT parameters $\epsilon^{IJ}{}_{\mu\nu}(x)$ as

$$\delta_0 A^I{}_\mu = \epsilon^{IJ}{}_{\mu\nu} \frac{\delta S}{\delta A^J{}_\nu}. \tag{20}$$

The requirement that the variation of the action vanishes:

$$\delta S = \frac{\delta S}{\delta A^I{}_\mu} \frac{\delta S}{\delta A^J{}_\nu} \epsilon^{IJ}{}_{\mu\nu} = 0,$$

enforces the antisymmetry restriction on the HT parameters:

$$\epsilon^{IJ}{}_{\mu\nu} = -\epsilon^{JI}{}_{\nu\mu}.$$

Note that this equation can be satisfied in two different ways—the parameters can be either antisymmetric with respect to group indices $IJ$ and symmetric with respect to spacetime

indices $\mu\nu$, or vice versa. We, therefore, have two possible choices for their symmetry properties. The first possibility is defined as

$$\epsilon^{IJ}{}_{\mu\nu} = \epsilon^{IJ}{}_{\nu\mu} = -\epsilon^{JI}{}_{\mu\nu} = -\epsilon^{JI}{}_{\nu\mu}\,, \tag{21}$$

while the second possibility is defined as

$$\epsilon^{IJ}{}_{\mu\nu} = \epsilon^{JI}{}_{\mu\nu} = -\epsilon^{IJ}{}_{\nu\mu} = -\epsilon^{JI}{}_{\nu\mu}\,. \tag{22}$$

Varying the action, one obtains an explicit form of the HT transformation:

$$\delta_0 A^I{}_\mu = \epsilon^{IJ}{}_{\mu\nu}\varepsilon^{\nu\rho\sigma}\left(\partial_\rho A_{J\sigma} - \partial_\sigma A_{J\rho} + f_{KLJ}\,A^K{}_\rho A^L{}_\sigma\right). \tag{23}$$

In order to demonstrate that HT transformations have highly nontrivial implications, despite being trivial on-shell, it is instructive to discuss diffeomorphisms. Namely, looking at the action (15), one expects that the theory has diffeomorphism symmetry, since it is formulated in a manifestly covariant way using differential forms. However, one can check that diffeomorphisms are not a subgroup of the ordinary gauge symmetry group $\mathcal{G}_{CS}$ given by (18), but nevertheless can be obtained as a subgroup of the total gauge group (19). In other words, one can demonstrate that

$$Diff(\mathcal{M}_3) \not\subset \mathcal{G}_{CS}\,, \qquad \text{but} \qquad Diff(\mathcal{M}_3) \subset \mathcal{G}_{\text{total}} = \mathcal{G}_{CS} \ltimes \mathcal{G}_{HT}\,.$$

Let us examine this in detail. The diffeomorphism transformation

$$x^\mu \to x'^\mu = x^\mu + \xi^\mu(x)\,, \tag{24}$$

determined by the parameter $\xi^\mu(x)$ represents a subgroup $Diff(\mathcal{M})$ of the full gauge symmetry of some given action, if for every field $\phi(x)$ in the theory and every choice of diffeomorphism parameters $\xi^\mu(x)$, there exists a choice of the gauge parameters $\epsilon^{\text{gauge}}(x)$ and the HT parameters $\epsilon^{\text{HT}}(x)$, such that:

$$\delta_0{}^{\text{diff}}\phi = \delta_0{}^{\text{gauge}}\phi + \delta_0{}^{\text{HT}}\phi\,. \tag{25}$$

In other words, if a theory has diffeomorphism symmetry, the diffeomorphism form variations of all the fields in the theory should be expressible in terms of their ordinary gauge and HT form variations.

In the case of Chern–Simons theory, this can be demonstrated explicitly. If one chooses the gauge parameters $\epsilon_{\mathfrak{g}}{}^I$ and the HT parameters $\epsilon^{IJ}{}_{\mu\nu}$ as

$$\epsilon_{\mathfrak{g}}{}^I = -\xi^\lambda A^I{}_\lambda\,, \qquad \epsilon^{IJ}{}_{\mu\nu} = -\frac{1}{2}\xi^\lambda \varepsilon_{\lambda\mu\nu} g^{IJ}\,, \tag{26}$$

where $g^{IJ}$ is the inverse of $g_{IJ} \equiv \langle T_I, T_J \rangle_{\mathfrak{g}}$, one can apply Equations (25) using (17) and (23) to reproduce precisely the well-known diffeomorphism form variation of the connection $A^I{}_\mu$:

$$\delta_0{}^{\text{diff}} A^I{}_\mu = -A^I{}_\lambda \partial_\mu \xi^\lambda - \xi^\lambda \partial_\lambda A^I{}_\mu\,. \tag{27}$$

Therefore, as expected, despite the fact that $Diff(\mathcal{M}_3) \not\subset \mathcal{G}_{CS}$, one obtains that $Diff(\mathcal{M}_3) \subset \mathcal{G}_{\text{total}} = \mathcal{G}_{CS} \ltimes \mathcal{G}_{HT}$. Note that the choice of HT parameters in (26) is nontrivial, which emphasizes the role of HT transformations and the fact that the full group of gauge symmetries is $\mathcal{G}_{\text{total}}$ rather than $\mathcal{G}_{CS}$. As we shall see in the next section, this property is not specific only to the Chern–Simons theory.

## 4. HT Symmetry in 3BF Theory

After discussing the Chern–Simons theory as a toy example, we move to the more important case of the 3BF theory. This theory is relevant for building models of quantum

gravity; see [8,20,21,33,35]. Therefore, it is important to study its gauge symmetry and, in particular, the role of HT transformations.

*4.1. Review of the 3BF Theory*

Analogous to the fact that Chern–Simons theory is a topological theory based on a Lie group and a 3-dimensional manifold, the $3BF$ theory is also a topological theory based on a notion of a three-group and a 4-dimensional manifold. The notion of a three-group represents a categorical generalization of the notion of a group, in the context of higher gauge theory (HGT); see [15] for a review and motivation. For the purpose of defining the $3BF$ theory, we are interested in particular in a strict Lie three-group, which is known to be isomorphic to a so-called Lie two-crossed module; see [17–19] for details.

A Lie two-crossed module, denoted as $(L \xrightarrow{\delta} H \xrightarrow{\partial} G, \rhd, \{\_,\_\}_{\mathrm{pf}})$, is an algebraic structure specified by three Lie groups $G$, $H$, and $L$, together with the homomorphisms $\delta : L \to H$ and $\partial : H \to G$, an action $\rhd$ of the group $G$ on all three groups, and a $G$-equivariant map, called the Peiffer lifting:

$$\{\_,\_\}_{\mathrm{pf}} : H \times H \to L.$$

In order for this structure to form a two-crossed module, the structure constants of algebras $\mathfrak{g}$, $\mathfrak{h}$, and $\mathfrak{l}$ (the Lie algebras corresponding to the Lie groups $G$, $H$, and $L$, respectively), as well as the maps $\partial$ and $\delta$, the action $\rhd$, and the Peiffer lifting, must satisfy certain axioms; see [20] for details.

Given a two-crossed module and a four-dimensional compact and orientable spacetime manifold $\mathcal{M}_4$, one can introduce the notion of a trivial principal three-bundle, in analogy with the notion of a trivial principal bundle constructed from an ordinary Lie group and a manifold; see [15]. Then, one can introduce the notion of a three-connection, an ordered triple $(\alpha, \beta, \gamma)$, where $\alpha$, $\beta$, and $\gamma$ are algebra-valued differential forms, $\alpha \in \mathcal{A}^1(\mathcal{M}_4, \mathfrak{g})$, $\beta \in \mathcal{A}^2(\mathcal{M}_4, \mathfrak{h})$, and $\gamma \in \mathcal{A}^3(\mathcal{M}_4, \mathfrak{l})$; see [17–19]. The corresponding fake hree-curvature $(\mathcal{F}, \mathcal{G}, \mathcal{H})$ is defined as:

$$\mathcal{F} = \mathrm{d}\alpha + \alpha \wedge \alpha - \partial\beta, \qquad \mathcal{G} = \mathrm{d}\beta + \alpha \wedge^{\rhd} \beta - \delta\gamma,$$
$$\mathcal{H} = \mathrm{d}\gamma + \alpha \wedge^{\rhd} \gamma + \{\beta \wedge \beta\}_{\mathrm{pf}}. \tag{28}$$

Then, for a four-dimensional manifold $\mathcal{M}_4$, one can define the gauge-invariant topological $3BF$ action, based on the structure of a two-crossed module $(L \xrightarrow{\delta} H \xrightarrow{\partial} G, \rhd, \{\_,\_\}_{\mathrm{pf}})$, by the action

$$S_{3BF} = \int_{\mathcal{M}_4} \langle B \wedge \mathcal{F} \rangle_{\mathfrak{g}} + \langle C \wedge \mathcal{G} \rangle_{\mathfrak{h}} + \langle D \wedge \mathcal{H} \rangle_{\mathfrak{l}}, \tag{29}$$

where $B \in \mathcal{A}^2(\mathcal{M}_4, \mathfrak{g})$, $C \in \mathcal{A}^1(\mathcal{M}_4, \mathfrak{h})$, and $D \in \mathcal{A}^0(\mathcal{M}_4, \mathfrak{l})$ are Lagrange multipliers and $\mathcal{F} \in \mathcal{A}^2(\mathcal{M}_4, \mathfrak{g})$, $\mathcal{G} \in \mathcal{A}^3(\mathcal{M}_4, \mathfrak{h})$, and $\mathcal{H} \in \mathcal{A}^4(\mathcal{M}_4, \mathfrak{l})$ represent the fake three-curvature given by Equation (28). The forms $\langle \_,\_ \rangle_{\mathfrak{g}}$, $\langle \_,\_ \rangle_{\mathfrak{h}}$, and $\langle \_,\_ \rangle_{\mathfrak{l}}$ are $G$-invariant symmetric nondegenerate bilinear forms on $\mathfrak{g}$, $\mathfrak{h}$, and $\mathfrak{l}$, respectively. The action (29) is an example of the so-called higher gauge theory.

By choosing the three bases of generators $\tau_\alpha \in \mathfrak{g}$, $t_a \in \mathfrak{h}$, and $T_A \in \mathfrak{l}$ of the three respective Lie algebras, one can expand all fields in the theory into components as

$$
\begin{aligned}
B &= \frac{1}{2} B^\alpha{}_{\mu\nu}(x)\, \mathrm{d}x^\mu \wedge \mathrm{d}x^\nu \otimes \tau_\alpha, & \alpha &= \alpha^\alpha{}_\mu(x)\, \mathrm{d}x^\mu \otimes \tau_\alpha, \\
C &= C^a{}_\mu(x)\, \mathrm{d}x^\mu \otimes t_a, & \beta &= \frac{1}{2} \beta^a{}_{\mu\nu}(x)\, \mathrm{d}x^\mu \wedge \mathrm{d}x^\nu \otimes t_a, \\
D &= D^A(x) T_A, & \gamma &= \frac{1}{3!} \gamma^A{}_{\mu\nu\rho}(x)\, \mathrm{d}x^\mu \wedge \mathrm{d}x^\nu \wedge \mathrm{d}x^\rho \otimes T_A.
\end{aligned}
$$

One can also make use of the following notation for the components of all maps present in the theory, in the same three bases:

$$[\tau_\alpha, \tau_\beta] = f_{\alpha\beta}{}^\gamma \tau_\gamma, \qquad g_{\alpha\beta} = \langle \tau_\alpha, \tau_\beta \rangle_{\mathfrak{g}}, \qquad \tau_\alpha \triangleright \tau_\beta = \triangleright_{\alpha\beta}{}^\gamma \tau_\gamma, \qquad \delta T_A = \delta_A{}^a t_a,$$

$$[t_a, t_b] = f_{ab}{}^c t_c, \qquad g_{ab} = \langle t_a, t_b \rangle_{\mathfrak{h}}, \qquad \tau_\alpha \triangleright t_a = \triangleright_{\alpha a}{}^b t_b, \qquad \partial t_a = \partial_a{}^\alpha \tau_\alpha,$$

$$[T_A, T_B] = f_{AB}{}^C T_C, \quad g_{AB} = \langle T_A, T_B \rangle_{\mathfrak{l}}, \quad \tau_\alpha \triangleright T_A = \triangleright_{\alpha A}{}^B T_B, \quad \{t_a, t_b\}_{\mathrm{pf}} = X_{ab}{}^A T_A.$$

The complete gauge symmetry of the $3BF$ action was studied in [8] using the techniques of Hamiltonian analysis. It consists of five types of gauge transformations, $G$-, $H$-, $L$-, $M$-, and $N$-gauge transformations, determined with the independent parameters $\epsilon_{\mathfrak{g}}{}^\alpha(x)$, $\epsilon_{\mathfrak{h}}{}^a{}_\mu(x)$, $\epsilon_{\mathfrak{l}}{}^A{}_{\mu\nu}(x)$, $\epsilon_{\mathfrak{m}}{}^\alpha{}_\mu(x)$, and $\epsilon_{\mathfrak{n}}{}^a(x)$, respectively. The form variations of the fields $B$, $C$, $D$, $\alpha$, $\beta$, and $\gamma$, obtained in [8] are given as follows:

$$
\begin{aligned}
\delta_0 B^\alpha{}_{\mu\nu} &= f_{\beta\gamma}{}^\alpha \epsilon_{\mathfrak{g}}{}^\beta B^\gamma{}_{\mu\nu} + 2 C_{a[\mu|} \epsilon_{\mathfrak{h}}{}^b{}_{|\nu]} \triangleright_{\beta b}{}^a g^{\alpha\beta} - D_A \triangleright_{\beta B}{}^A \epsilon_{\mathfrak{l}}{}^B{}_{\mu\nu} g^{\alpha\beta} - 2 \nabla_{[\mu|} \epsilon_{\mathfrak{m}}{}^\alpha{}_{|\nu]} \\
&\quad + \beta_{b\mu\nu} \triangleright_{\beta a}{}^b \epsilon_{\mathfrak{n}}{}^a g^{\alpha\beta}, \\
\delta_0 C^a{}_\mu &= \triangleright_{\alpha b}{}^a \epsilon_{\mathfrak{g}}{}^\alpha C^b{}_\mu + 2 D_A X_{(ab)}{}^A \epsilon_{\mathfrak{h}}{}^b{}_\mu - \partial^a{}_\alpha \epsilon_{\mathfrak{m}}{}^\alpha{}_\mu - \nabla_\mu \epsilon_{\mathfrak{n}}{}^a, \\
\delta_0 D^A &= \triangleright_{\alpha B}{}^A \epsilon_{\mathfrak{g}}{}^\alpha D^B + \delta^A{}_a \epsilon_{\mathfrak{n}}{}^a, \\
\delta_0 \alpha^\alpha{}_\mu &= -\partial_\mu \epsilon_{\mathfrak{g}}{}^\alpha - f_{\beta\gamma}{}^\alpha \alpha^\beta{}_\mu \epsilon_{\mathfrak{g}}{}^\gamma - \partial_a{}^\alpha \epsilon_{\mathfrak{h}}{}^a{}_\mu, \\
\delta_0 \beta^a{}_{\mu\nu} &= \triangleright_{\alpha b}{}^a \epsilon_{\mathfrak{g}}{}^\alpha \beta^b{}_{\mu\nu} - 2 \nabla_{[\mu|} \epsilon_{\mathfrak{h}}{}^a{}_{|\nu]} + \delta_A{}^a \epsilon_{\mathfrak{l}}{}^A{}_{\mu\nu}, \\
\delta_0 \gamma^A{}_{\mu\nu\rho} &= \triangleright_{\alpha B}{}^A \epsilon_{\mathfrak{g}}{}^\alpha \gamma^B{}_{\mu\nu\rho} + 3! \beta^a{}_{[\mu\nu} \epsilon_{\mathfrak{h}}{}^b{}_{\rho]} X_{(ab)}{}^A + \nabla_\mu \epsilon_{\mathfrak{l}}{}^A{}_{\nu\rho} - \nabla_\nu \epsilon_{\mathfrak{l}}{}^A{}_{\mu\rho} + \nabla_\rho \epsilon_{\mathfrak{l}}{}^A{}_{\mu\nu}.
\end{aligned}
\tag{30}
$$

The gauge transformations (30) form a group $\mathcal{G}_{3BF}$:

$$\mathcal{G}_{3BF} = \tilde{G} \ltimes (\tilde{H}_L \ltimes (\tilde{N} \times \tilde{M})), \tag{31}$$

where $\tilde{G}$ denotes the group of $G$-gauge transformations, the $H$-gauge transformations together with the $L$-gauge transformations form the group $\tilde{H}_L$, while $\tilde{M}$ and $\tilde{N}$ are the groups of $M$- and $N$-gauge transformations, respectively. All these groups are determined from the structure of the initial chosen two-crossed module that defines the theory; see [8] for details.

However, as we have seen in the general theory in Section 2 and in the example of the Chern–Simons theory in Section 3, the symmetry group $\mathcal{G}_{3BF}$ determined by the Hamiltonian analysis does not include HT transformations, and therefore, the *total* gauge group should in fact be

$$\mathcal{G}_{\mathrm{total}} = \mathcal{G}_{3BF} \ltimes \mathcal{G}_{HT}. \tag{32}$$

*4.2. Explicit HT Transformations*

Let us explicitly define the *HT* transformations for the $3BF$ action (29). If we denote the dimensions of the Lie algebras $\mathfrak{g}, \mathfrak{h}, \mathfrak{l}$ as

$$\dim(\mathfrak{g}) = p, \qquad \dim(\mathfrak{h}) = q, \qquad \dim(\mathfrak{l}) = r,$$

the number of independent field components in the theory can be counted according to the following table:

| $B^\alpha{}_{\mu\nu}$ | $C^a{}_\mu$ | $D^A$ | $\alpha^\alpha{}_\mu$ | $\beta^a{}_{\mu\nu}$ | $\gamma^A{}_{\mu\nu\rho}$ |
|---|---|---|---|---|---|
| $6p$ | $4q$ | $r$ | $4p$ | $6q$ | $4r$ |

The total number of independent field components is, therefore,

$$N = 6p + 4q + r + 4p + 6q + 4r = 10p + 10q + 5r.$$

Let $\phi^i$ denote all field components, where $i = 1, 2, \ldots, N$. We can write the fields schematically as a column-matrix with six blocks:

$$\phi^i = \begin{pmatrix} B^\alpha{}_{\mu\nu} \\ C^a{}_\mu \\ D^A \\ \alpha^\alpha{}_\mu \\ \beta^a{}_{\mu\nu} \\ \gamma^A{}_{\mu\nu\rho} \end{pmatrix}.$$

The HT transformation is then defined via the parameters $\epsilon^{ij}(x)$ as

$$\delta_0 \phi^i = \epsilon^{ij} \frac{\delta S}{\delta \phi^j}.$$

The requirement that the variation of the action vanishes enforces the antisymmetry restriction on the parameters, $\epsilon^{ij} = -\epsilon^{ji}$, for all $i, j \in \{1, \ldots, N\}$. These transformations can be represented more explicitly as a tensorial $6 \times 6$ block-matrix equation, in the following form:

$$\begin{pmatrix} \delta_0 B^\alpha{}_{\mu\nu} \\ \delta_0 C^a{}_\mu \\ \delta_0 D^A \\ \delta_0 \alpha^\alpha{}_\mu \\ \delta_0 \beta^a{}_{\mu\nu} \\ \delta_0 \gamma^A{}_{\mu\nu\rho} \end{pmatrix} = \begin{pmatrix} \epsilon^{\alpha\beta}{}_{\mu\nu\sigma\lambda} & \epsilon^{\alpha b}{}_{\mu\nu\sigma} & \epsilon^{\alpha B}{}_{\mu\nu} & \epsilon^{\alpha\beta}{}_{\mu\nu\sigma} & \epsilon^{\alpha b}{}_{\mu\nu\sigma\lambda} & \epsilon^{\alpha B}{}_{\mu\nu\sigma\lambda\xi} \\ \mu^{a\beta}{}_{\mu\sigma\lambda} & \epsilon^{ab}{}_{\mu\sigma} & \epsilon^{aB}{}_\mu & \epsilon^{a\beta}{}_{\mu\sigma} & \epsilon^{ab}{}_{\mu\sigma\lambda} & \epsilon^{aB}{}_{\mu\sigma\lambda\xi} \\ \mu^{A\beta}{}_{\sigma\lambda} & \mu^{Ab}{}_\sigma & \epsilon^{AB} & \epsilon^{A\beta}{}_\sigma & \epsilon^{Ab}{}_{\sigma\lambda} & \epsilon^{AB}{}_{\sigma\lambda\xi} \\ \mu^{\alpha\beta}{}_{\mu\sigma\lambda} & \mu^{\alpha b}{}_{\mu\sigma} & \mu^{\alpha B}{}_\mu & \epsilon^{\alpha\beta}{}_{\mu\sigma} & \epsilon^{\alpha b}{}_{\mu\sigma\lambda} & \epsilon^{\alpha B}{}_{\mu\sigma\lambda\xi} \\ \mu^{a\beta}{}_{\mu\nu\sigma\lambda} & \mu^{ab}{}_{\mu\nu\sigma} & \mu^{aB}{}_{\mu\nu} & \mu^{a\beta}{}_{\mu\nu\sigma} & \epsilon^{ab}{}_{\mu\nu\sigma\lambda} & \epsilon^{aB}{}_{\mu\nu\sigma\lambda\xi} \\ \mu^{A\beta}{}_{\mu\nu\rho\sigma\lambda} & \mu^{Ab}{}_{\mu\nu\rho\sigma} & \mu^{AB}{}_{\mu\nu\rho} & \mu^{A\beta}{}_{\mu\nu\rho\sigma} & \mu^{Ab}{}_{\mu\nu\rho\sigma\lambda} & \epsilon^{AB}{}_{\mu\nu\rho\sigma\lambda\xi} \end{pmatrix} \begin{pmatrix} \frac{1}{2}\frac{\delta S}{\delta B^\beta{}_{\sigma\lambda}} \\ \frac{\delta S}{\delta C^b{}_\sigma} \\ \frac{\delta S}{\delta D^B} \\ \frac{\delta S}{\delta \alpha^\beta{}_\sigma} \\ \frac{1}{2}\frac{\delta S}{\delta \beta^b{}_{\sigma\lambda}} \\ \frac{1}{3!}\frac{\delta S}{\delta \gamma^B{}_{\sigma\lambda\xi}} \end{pmatrix}. \quad (33)$$

The coefficients multiplying the variations of the action in the column on the right-hand side are there to compensate the overcounting of the independent field components. Due to the antisymmetry of HT parameters, all $\mu$ blocks (below the diagonal) are determined in terms of the $\epsilon$ blocks (above the diagonal), as follows. For the first column of the parameter matrix in (33), we have:

$$\mu^{b\alpha}{}_{\sigma\mu\nu} = -\epsilon^{\alpha b}{}_{\mu\nu\sigma}, \qquad \mu^{B\alpha}{}_{\mu\nu} = -\epsilon^{\alpha B}{}_{\mu\nu}, \qquad \mu^{\beta\alpha}{}_{\sigma\mu\nu} = -\epsilon^{\alpha\beta}{}_{\mu\nu\sigma},$$
$$\mu^{b\alpha}{}_{\sigma\lambda\mu\nu} = -\epsilon^{\alpha b}{}_{\mu\nu\sigma\lambda}, \qquad \mu^{B\alpha}{}_{\sigma\lambda\xi\mu\nu} = -\epsilon^{\alpha B}{}_{\mu\nu\sigma\lambda\xi}. \quad (34)$$

For the second column, we have:

$$\mu^{Ba}{}_\mu = -\epsilon^{aB}{}_\mu, \qquad \mu^{\beta a}{}_{\sigma\mu} = -\epsilon^{a\beta}{}_{\mu\sigma},$$
$$\mu^{ba}{}_{\sigma\lambda\mu} = -\epsilon^{ab}{}_{\mu\sigma\lambda}, \qquad \mu^{Ba}{}_{\sigma\lambda\xi\mu} = -\epsilon^{aB}{}_{\mu\sigma\lambda\xi}. \quad (35)$$

The $\mu$ parameters in the third column are determined via:

$$\mu^{\beta A}{}_\sigma = -\epsilon^{A\beta}{}_\sigma, \qquad \mu^{bA}{}_{\sigma\lambda} = -\epsilon^{Ab}{}_{\sigma\lambda}, \qquad \mu^{BA}{}_{\sigma\lambda\xi} = -\epsilon^{AB}{}_{\sigma\lambda\xi}, \quad (36)$$

while the remaining $\mu$ parameters in the fourth and fifth columns are determined as:

$$\mu^{b\alpha}{}_{\sigma\lambda\mu} = -\epsilon^{\alpha b}{}_{\mu\sigma\lambda}, \qquad \mu^{B\alpha}{}_{\sigma\lambda\xi\mu} = -\epsilon^{\alpha B}{}_{\mu\sigma\lambda\xi}, \qquad \mu^{Ba}{}_{\sigma\lambda\xi\mu\nu} = -\epsilon^{aB}{}_{\mu\nu\sigma\lambda\xi}. \quad (37)$$

Finally, in addition to all these, the parameters in the blocks on the diagonal also have to satisfy certain antisymmetry relations, specifically:

$$\epsilon^{\alpha\beta}{}_{\mu\nu\sigma\lambda} = -\epsilon^{\beta\alpha}{}_{\sigma\lambda\mu\nu}, \qquad \epsilon^{ab}{}_{\mu\sigma} = -\epsilon^{ba}{}_{\sigma\mu}, \qquad \epsilon^{AB} = -\epsilon^{BA},$$

$$\epsilon^{\alpha\beta}{}_{\mu\sigma} = -\epsilon^{\beta\alpha}{}_{\sigma\mu}, \qquad \epsilon^{ab}{}_{\mu\nu\sigma\lambda} = -\epsilon^{ba}{}_{\sigma\lambda\mu\nu}, \qquad \epsilon^{AB}{}_{\mu\nu\rho\sigma\lambda\xi} = -\epsilon^{BA}{}_{\sigma\lambda\xi\mu\nu\rho}. \tag{38}$$

Like in the example of the Chern–Simons theory from the previous section, these antisymmetry relations can be satisfied in various multiple ways. All those possibilities are allowed, as long as the identities (38) are satisfied. The final ingredient in (33) is the expressions for the variation of the action with respect to the fields, and these are given as follows:

$$
\begin{aligned}
\frac{\delta S}{\delta B^{\beta}{}_{\nu\rho}} &= \frac{1}{2}\varepsilon^{\nu\rho\sigma\tau}\mathcal{F}_{\beta\sigma\tau}, \\
\frac{\delta S}{\delta C^{b}{}_{\rho}} &= \frac{1}{3!}\varepsilon^{\rho\sigma\tau\lambda}\mathcal{G}_{b\sigma\tau\lambda}, \\
\frac{\delta S}{\delta D^{B}} &= \frac{1}{4!}\varepsilon^{\sigma\tau\lambda\xi}\mathcal{H}_{B\sigma\tau\lambda\xi}, \\
\frac{\delta S}{\delta\alpha^{\beta}{}_{\rho}} &= \frac{1}{2}\varepsilon^{\rho\tau\lambda\xi}\left(\nabla_{\tau}B_{\beta\lambda\xi} - \triangleright_{\beta a}{}^{b}C_{b\tau}\beta^{a}{}_{\lambda\xi} + \frac{1}{3}\triangleright_{\beta B}{}^{A}D_{A}\gamma^{B}{}_{\tau\lambda\xi}\right), \\
\frac{\delta S}{\delta\beta^{b}{}_{\nu\rho}} &= \varepsilon^{\nu\rho\sigma\tau}\left(\nabla_{\sigma}C_{b\tau} - \frac{1}{2}\partial_{b}{}^{\alpha}B_{\alpha\sigma\tau} + X_{(ab)}{}^{A}D_{A}\beta^{b}{}_{\sigma\tau}\right), \\
\frac{\delta S}{\delta\gamma^{B}{}_{\mu\nu\rho}} &= \varepsilon^{\mu\nu\rho\sigma}(\nabla_{\sigma}D_{B} + \delta_{B}{}^{a}C_{a\sigma}).
\end{aligned}
\tag{39}
$$

*4.3. Diffeomorphisms*

As in the case of the Chern–Simons theory, it is instructive to discuss diffeomorphism symmetry. The $3BF$ action (29) obviously is diffeomorphism invariant, since it is formulated in a manifestly covariant way, using differential forms. However, one can check that the diffeomorphisms are not a subgroup of the gauge symmetry group $\mathcal{G}_{3BF}$ given by Equation (31), but nevertheless can be obtained as a subgroup of the total gauge group (32):

$$Diff(\mathcal{M}_4) \not\subset \mathcal{G}_{3BF}, \qquad \text{but} \qquad Diff(\mathcal{M}_4) \subset \mathcal{G}_{\text{total}} = \mathcal{G}_{3BF} \ltimes \mathcal{G}_{HT}. \tag{40}$$

Let us demonstrate this. Like in the Chern–Simons case, we want to demonstrate that the form variation of all fields corresponding to diffeomorphisms can be obtained as a suitable combination of the form variations for the ordinary gauge transformations (30) and the HT transformations (33). In other words, for an arbitrary choice of the diffeomorphism parameters $\xi^{\mu}(x)$ from (24), Equation (25) should hold in the case of the $3BF$ theory as well:

$$\delta_0{}^{\text{diff}}\phi = \delta_0{}^{\text{gauge}}\phi + \delta_0{}^{\text{HT}}\phi. \tag{41}$$

Indeed, this can be shown by a suitable choice of parameters. Regarding the parameters of the gauge transformations (30), the appropriate choice is given as:

$$
\begin{aligned}
\epsilon_{\mathfrak{g}}{}^{\alpha} &= \xi^{\lambda}\alpha^{\alpha}{}_{\lambda}, \qquad \epsilon_{\mathfrak{h}}{}^{a}{}_{\mu} = -\xi^{\lambda}\beta^{a}{}_{\mu\lambda}, \qquad \epsilon_{\mathfrak{l}}{}^{A}{}_{\mu\nu} = -\xi^{\lambda}\gamma^{A}{}_{\mu\nu\lambda}, \\
\epsilon_{\mathfrak{m}}{}^{\alpha}{}_{\mu} &= -\xi^{\lambda}B^{\alpha}{}_{\mu\lambda}, \qquad \epsilon_{\mathfrak{n}}{}^{a} = \xi^{\lambda}C^{a}{}_{\lambda}.
\end{aligned}
\tag{42}
$$

Regarding the parameters of the HT transformations (33), we chose the following special case, with the majority of the parameters equated to zero:

$$
\begin{pmatrix}
\delta_0 B^\alpha{}_{\mu\nu} \\
\delta_0 C^a{}_\mu \\
\delta_0 D^A \\
\delta_0 \alpha^\alpha{}_\mu \\
\delta_0 \beta^a{}_{\mu\nu} \\
\delta_0 \gamma^A{}_{\mu\nu\rho}
\end{pmatrix}
=
\begin{pmatrix}
0 & 0 & 0 & \epsilon^{\alpha\beta}{}_{\mu\nu\sigma} & 0 & 0 \\
0 & 0 & 0 & 0 & \epsilon^{ab}{}_{\mu\sigma\lambda} & 0 \\
0 & 0 & 0 & 0 & 0 & \epsilon^{AB}{}_{\sigma\lambda\xi} \\
\mu^{\alpha\beta}{}_{\mu\sigma\lambda} & 0 & 0 & 0 & 0 & 0 \\
0 & \mu^{ab}{}_{\mu\nu\sigma} & 0 & 0 & 0 & 0 \\
0 & 0 & \mu^{AB}{}_{\mu\nu\rho} & 0 & 0 & 0
\end{pmatrix}
\begin{pmatrix}
\frac{1}{2}\frac{\delta S}{\delta B^\beta{}_{\sigma\lambda}} \\
\frac{\delta S}{\delta C^b{}_\sigma} \\
\frac{\delta S}{\delta D^B} \\
\frac{\delta S}{\delta \alpha^\beta{}_\sigma} \\
\frac{1}{2}\frac{\delta S}{\delta \beta^b{}_{\sigma\lambda}} \\
\frac{1}{3!}\frac{\delta S}{\delta \gamma^B{}_{\sigma\lambda\xi}}
\end{pmatrix}.
\tag{43}
$$

Of course, due to antisymmetry, the nonzero $\mu$ blocks take negative values of the corresponding $\epsilon$ blocks, in accordance with (34), (35), and (36). The three independent nonzero $\epsilon$ blocks are chosen as

$$
\epsilon^{\alpha\beta}{}_{\mu\nu\sigma} = \xi^\rho g^{\alpha\beta}\varepsilon_{\mu\nu\sigma\rho}\,, \qquad
\epsilon^{ab}{}_{\mu\sigma\lambda} = \xi^\rho g^{ab}\varepsilon_{\rho\mu\sigma\lambda}\,, \qquad
\epsilon^{AB}{}_{\sigma\lambda\xi} = \xi^\rho g^{AB}\varepsilon_{\sigma\lambda\xi\rho}\,.
\tag{44}
$$

Finally, substituting (42) and (44) into (30) and (43), respectively, and then substituting all those results into (41), after a certain amount of work, one obtains precisely the standard form variations corresponding to diffeomorphisms:

$$
\begin{aligned}
\delta_0^{\text{diff}} B^\alpha{}_{\mu\nu} &= -B^\alpha{}_{\lambda\nu}\partial_\mu\xi^\lambda - B^\alpha{}_{\mu\lambda}\partial_\nu\xi^\lambda - \xi^\lambda\partial_\lambda B^\alpha{}_{\mu\nu}\,, \\
\delta_0^{\text{diff}} C^a{}_\mu &= -C^a{}_\lambda\partial_\mu\xi^\lambda - \xi^\lambda\partial_\lambda C^a{}_\mu\,, \\
\delta_0^{\text{diff}} D^A &= -\xi^\lambda\partial_\lambda D^A\,, \\
\delta_0^{\text{diff}} \alpha^\alpha{}_\mu &= -\alpha^\alpha{}_\lambda\partial_\mu\xi^\lambda - \xi^\lambda\partial_\lambda\alpha^\alpha{}_\mu\,, \\
\delta_0^{\text{diff}} \beta^a{}_{\mu\nu} &= -\beta^a{}_{\lambda\nu}\partial_\mu\xi^\lambda - \beta^a{}_{\mu\lambda}\partial_\nu\xi^\lambda - \xi^\lambda\partial_\lambda\beta^a{}_{\mu\nu}\,, \\
\delta_0^{\text{diff}} \gamma^A{}_{\mu\nu\rho} &= -\gamma^A{}_{\lambda\nu\rho}\partial_\mu\xi^\lambda - \gamma^A{}_{\mu\lambda\rho}\partial_\nu\xi^\lambda - \gamma^A{}_{\mu\nu\lambda}\partial_\rho\xi^\lambda - \xi^\lambda\partial_\lambda\gamma^A{}_{\mu\nu\rho}\,.
\end{aligned}
\tag{45}
$$

This establishes both relations (40), as we set out to demonstrate. We note again that the HT transformations play a crucial role in obtaining the result, since we had to choose the parameters (44) in a nontrivial manner.

*4.4. Symmetry Breaking in 2BF Theory*

Let us now turn to the topic of symmetry breaking and the way it influences HT transformations. To that end, we studied the topological $2BF$ action, which is a special case of the $3BF$ action (29) without the last term:

$$
S_{2BF} = \int_{\mathcal{M}_4} \langle B \wedge \mathcal{F}\rangle_\mathfrak{g} + \langle C \wedge \mathcal{G}\rangle_\mathfrak{h}\,.
\tag{46}
$$

In order to be even more concrete, let us fix a two-crossed module structure with the following choice of groups:

$$
G = SO(3,1)\,, \qquad H = \mathbb{R}^4\,, \qquad L = \{e\}\,.
$$

In other words, we interpret group $G$ as the Lorentz group, group $H$ as the spacetime translations group, while group $L$ is trivial, for simplicity. This choice corresponds to the so-called Poincaré two-group; see [16] for details. Since the generators of the Lorentz group can be conveniently counted using the antisymmetric combinations of indices from the group of translations, instead of the $G$-group indices $\alpha$, we shall systematically write $[ab] \in \{01, 02, 03, 12, 13, 23\}$, where $a, b \in \{0, 1, 2, 3\}$ are $H$-group indices, and the brackets denote antisymmetrization. With a further change in notation from the connection 1-form $\alpha$ to the spin-connection 1-form $\omega$, the curvature 2-form $\mathcal{F}(\alpha)$ to $R(\omega)$, and interpreting

the Lagrange multiplier 1-form $C$ as the tetrad 1-form $e$, the $2BF$ action can be rewritten in new notation as

$$S_{2BF} = \int_{\mathcal{M}_4} B^{[ab]} \wedge R_{[ab]} + e^a \wedge \mathcal{G}_a \,. \tag{47}$$

The ordinary gauge symmetry group for this action has a form similar to (31):

$$\mathcal{G}_{2BF} = \tilde{G} \ltimes (\tilde{H} \ltimes (\tilde{N} \times \tilde{M})) \,, \tag{48}$$

while the total group of gauge symmetries is extended by the HT transformations, so that

$$\mathcal{G}_{\text{total}} = \mathcal{G}_{2BF} \ltimes \mathcal{G}_{HT} \,. \tag{49}$$

The explicit $HT$ transformations are written as a tensorial $4 \times 4$ block-matrix equation, in the form

$$\begin{pmatrix} \delta_0 B^{[ab]}{}_{\mu\nu} \\ \delta_0 e^a{}_\mu \\ \delta_0 \omega^{[ab]}{}_\mu \\ \delta_0 \beta^a{}_{\mu\nu} \end{pmatrix} = \begin{pmatrix} \epsilon^{[ab][cd]}{}_{\mu\nu\sigma\lambda} & \epsilon^{[ab]c}{}_{\mu\nu\sigma} & \epsilon^{[ab][cd]}{}_{\mu\nu\sigma} & \epsilon^{[ab]c}{}_{\mu\nu\sigma\lambda} \\ \mu^{a[cd]}{}_{\mu\sigma\lambda} & \epsilon^{ac}{}_{\mu\sigma} & \epsilon^{a[cd]}{}_{\mu\sigma} & \epsilon^{ac}{}_{\mu\sigma\lambda} \\ \mu^{[ab][cd]}{}_{\mu\sigma\lambda} & \mu^{[ab]c}{}_{\mu\sigma} & \epsilon^{[ab][cd]}{}_{\mu\sigma} & \epsilon^{[ab]c}{}_{\mu\sigma\lambda} \\ \mu^{a[cd]}{}_{\mu\nu\sigma\lambda} & \mu^{ac}{}_{\mu\nu\sigma} & \mu^{a[cd]}{}_{\mu\nu\sigma} & \epsilon^{ac}{}_{\mu\nu\sigma\lambda} \end{pmatrix} \begin{pmatrix} \frac{1}{4}\frac{\delta S}{\delta B^{[cd]}{}_{\sigma\lambda}} \\ \frac{\delta S}{\delta e^c{}_\sigma} \\ \frac{1}{2}\frac{\delta S}{\delta \omega^{[cd]}{}_\sigma} \\ \frac{1}{2}\frac{\delta S}{\delta \beta^c{}_{\sigma\lambda}} \end{pmatrix} \,, \tag{50}$$

where the usual antisymmetry rules apply. Here, we have

$$\begin{aligned} \frac{\delta S}{\delta B^{[cd]}{}_{\sigma\lambda}} &= \varepsilon^{\mu\nu\sigma\lambda} R_{[cd]\mu\nu} \,, \\ \frac{\delta S}{\delta \omega^{[cd]}{}_\sigma} &= \varepsilon^{\sigma\mu\nu\rho} \left( \nabla_\mu B_{[cd]\nu\rho} - e_{[c|\mu} \beta_{|d]\nu\rho} \right) \,, \\ \frac{\delta S}{\delta e^c{}_\sigma} &= \frac{1}{2} \varepsilon^{\sigma\mu\nu\rho} \nabla_\mu \beta_{c\nu\rho} \,, \\ \frac{\delta S}{\delta \beta^c{}_{\sigma\lambda}} &= \varepsilon^{\mu\nu\sigma\lambda} \nabla_\mu e_{c\nu} \,. \end{aligned} \tag{51}$$

The $2BF$ action (46) is topological, in the sense that it has no local propagating degrees of freedom. In this sense, it does not represent a theory of any realistic physics. In order to construct a more realistic theory, one proceeds by introducing the so-called *simplicity constraint* term into the action, which changes the equations of motion of the theory so that it does have nontrivial degrees of freedom. An example is the action

$$S_{GR} = \int_{\mathcal{M}_4} B^{[ab]} \wedge R_{[ab]} + e^a \wedge \nabla \beta_a - \lambda_{[ab]} \wedge \left( B^{[ab]} - \frac{1}{16\pi l_p^2} \varepsilon^{abcd} e_c \wedge e_d \right) \,, \tag{52}$$

where the new constraint term features another Lagrange multiplier two-form $\lambda_{[ab]}$. By virtue of the simplicity constraint, the theory becomes equivalent to general relativity, in the sense that the corresponding equations of motion reduce to vacuum Einstein field equations (see [16] for the analysis and proof). In this sense, constraint terms of various types are important when building more realistic theories; see [20] for more examples.

However, adding the simplicity constraint term also changes the gauge symmetry of the theory. In particular, it breaks the gauge group $\mathcal{G}_{2BF}$ from (48) down to one of its subgroups, so that the symmetry group of the action $S_{GR}$ is

$$\mathcal{G}_{GR} \subset \mathcal{G}_{2BF} \,. \tag{53}$$

This is expected and unsurprising. What is less obvious, however, is that the group of HT transformations $\tilde{\mathcal{G}}_{HT}$ of the action $S_{GR}$ *is not a subgroup* of the HT group $\mathcal{G}_{HT}$ of the original action $S_{2BF}$:

$$\tilde{\mathcal{G}}_{HT} \not\subset \mathcal{G}_{HT} \,, \tag{54}$$

which implies that

$$\mathcal{G}^{GR}_{\text{total}} \not\subset \mathcal{G}^{2BF}_{\text{total}}, \tag{55}$$

despite (53).

Let us demonstrate this. Since the action (52) features an additional field $\lambda^{[ab]}{}_{\mu\nu}(x)$, the HT transformations (50) have to be modified to take this into account and obtain the following $5 \times 5$ block-matrix form:

$$
\begin{pmatrix}
\delta_0 B^{[ab]}{}_{\mu\nu} \\
\delta_0 e^a{}_\mu \\
\delta_0 \omega^{[ab]}{}_\mu \\
\delta_0 \beta^a{}_{\mu\nu} \\
\delta_0 \lambda^{[ab]}{}_{\mu\nu}
\end{pmatrix}
=
\begin{pmatrix}
\epsilon^{[ab][cd]}{}_{\mu\nu\sigma\lambda} & \epsilon^{[ab]c}{}_{\mu\nu\sigma} & \epsilon^{[ab][cd]}{}_{\mu\nu\sigma} & \epsilon^{[ab]c}{}_{\mu\nu\sigma\lambda} & \zeta^{[ab][cd]}{}_{\mu\nu\sigma\xi} \\
\mu^{a[cd]}{}_{\mu\sigma\lambda} & \epsilon^{ac}{}_{\mu\sigma} & \epsilon^{a[cd]}{}_{\mu\sigma} & \epsilon^{ac}{}_{\mu\sigma\lambda} & \zeta^{a[cd]}{}_{\mu\sigma\xi} \\
\mu^{[ab][cd]}{}_{\mu\sigma\lambda} & \mu^{[ab]c}{}_{\mu\sigma} & \epsilon^{[ab][cd]}{}_{\mu\sigma} & \epsilon^{[ab]c}{}_{\mu\sigma\lambda} & \zeta^{[ab][cd]}{}_{\mu\sigma\xi} \\
\mu^{a[cd]}{}_{\mu\nu\sigma\lambda} & \mu^{ac}{}_{\mu\nu\sigma} & \mu^{a[cd]}{}_{\mu\nu\sigma} & \epsilon^{ac}{}_{\mu\nu\sigma\lambda} & \zeta^{a[cd]}{}_{\mu\nu\sigma\xi} \\
\theta^{[ab][cd]}{}_{\mu\nu\sigma\lambda} & \theta^{[ab]c}{}_{\mu\nu\sigma} & \theta^{[ab][cd]}{}_{\mu\nu\sigma} & \theta^{[ab]c}{}_{\mu\nu\sigma\lambda} & \psi^{[ab][cd]}{}_{\mu\nu\sigma\xi}
\end{pmatrix}
\begin{pmatrix}
\frac{1}{4} \frac{\delta S_{GR}}{\delta B^{[cd]}{}_{\sigma\lambda}} \\
\frac{\delta S_{GR}}{\delta e^c{}_\sigma} \\
\frac{1}{2} \frac{\delta S_{GR}}{\delta \omega^{[cd]}{}_\sigma} \\
\frac{1}{2} \frac{\delta S_{GR}}{\delta \beta^c{}_{\sigma\lambda}} \\
\frac{1}{4} \frac{\delta S_{GR}}{\delta \lambda^{[cd]}{}_{\sigma\xi}}
\end{pmatrix},
\tag{56}
$$

where

$$
\begin{aligned}
\frac{\delta S_{GR}}{\delta B^{[cd]}{}_{\sigma\lambda}} &= \varepsilon^{\mu\nu\sigma\lambda}\left( R_{[cd]\mu\nu} - \lambda_{[cd]\mu\nu} \right), \\
\frac{\delta S_{GR}}{\delta \omega^{[cd]}{}_\sigma} &= \varepsilon^{\sigma\mu\nu\rho}\left( \nabla_\mu B_{[cd]\nu\rho} - e_{[c|\mu}\beta_{|d]\nu\rho} \right), \\
\frac{\delta S_{GR}}{\delta e^c{}_\sigma} &= \frac{1}{2}\varepsilon^{\sigma\mu\nu\rho}\left( \nabla_\mu \beta_{c\nu\rho} + \frac{1}{8\pi l_p^2}\varepsilon_{abcd}\lambda^{[ab]}{}_{\mu\nu}e^d{}_\rho \right), \\
\frac{\delta S_{GR}}{\delta \beta^c{}_{\sigma\lambda}} &= \varepsilon^{\mu\nu\sigma\lambda}\nabla_\mu e_{c\nu}, \\
\frac{\delta S_{GR}}{\delta \lambda^{[cd]}{}_{\sigma\xi}} &= -\varepsilon^{\sigma\xi\mu\nu}\left( B_{[cd]\mu\nu} - \frac{1}{8\pi l_p^2}\varepsilon_{abcd}e^a{}_\mu e^b{}_\nu \right).
\end{aligned}
\tag{57}
$$

We can now investigate the differences in the form of HT transformations for the topological and constrained theory. First, comparing (56) to (50), we see that the HT transformations in the constrained theory feature *more gauge parameters* than are present in the topological theory. Namely, compared to $S_{2BF}$, the action $S_{GR}$ features an extra Lagrange multiplier two-form $\lambda^{[ab]}$, which extends the matrix of HT parameters from $4 \times 4$ blocks to $5 \times 5$ blocks, and, therefore, introduces the new parameters $\zeta$ and $\psi$ (and $\theta$, which are the negative of $\zeta$ due to antisymmetry). This means that the group $\tilde{\mathcal{G}}_{HT}$ for the constrained theory is *larger* than the group $\mathcal{G}_{HT}$ for the topological theory. On the one hand, this immediately proves (54) and, consequently, (55). On the other hand, one can ask the opposite question—given that $\tilde{\mathcal{G}}_{HT}$ is larger than $\mathcal{G}_{HT}$, is the latter maybe a subgroup of the former?

The answer to this question is negative:

$$\mathcal{G}_{HT} \not\subset \tilde{\mathcal{G}}_{HT}, \tag{58}$$

which together with (54) implies our final conclusion:

$$\mathcal{G}_{HT} \neq \tilde{\mathcal{G}}_{HT}. \tag{59}$$

In order to demonstrate (58), we can try to set all extra parameters $\zeta$, $\psi$, and $\theta$ to zero in (56), reducing it to the same form as (50). This would naively suggest that $\mathcal{G}_{HT}$ indeed is a subgroup of $\tilde{\mathcal{G}}_{HT}$. However, upon closer inspection, we can observe that this is not true, since the functional derivatives (57) are different from (51). Namely, even taking into account that the choice $\zeta = \psi = \theta = 0$ eliminates the fifth equation from (57), the first four equations are still different from their counterparts (51) because of the presence of the Lagrange multiplier $\lambda^{[ab]}$ in the action. The Lagrange multiplier is a field in the theory, and generically, it is not zero, since it is determined by the equation of motion:

$$\lambda^{[ab]}{}_{\mu\nu} = R^{[ab]}{}_{\mu\nu}.$$

Therefore, the HT transformations (56) in fact cannot be reduced to the HT transformations (50) by setting the extra parameters equal to zero, which proves (58) and (59).

The overall consequences from the above analysis are as follows. The topological action $S_{2BF}$ has a large ordinary gauge group $\mathcal{G}_{2BF}$ and a small HT symmetry group $\mathcal{G}_{HT}$. When one changes the action to $S_{GR}$ by adding a simplicity constraint term, two things happen—the ordinary gauge group breaks down to its subgroup $\mathcal{G}_{GR}$, so that it becomes smaller, while the HT symmetry group *grows larger* to a completely different group $\tilde{\mathcal{G}}_{HT}$. In effect, the *total* gauge groups for the two actions are intrinsically different:

$$\mathcal{G}_{\text{total}}^{2BF} = \mathcal{G}_{2BF} \ltimes \mathcal{G}_{HT} \qquad \neq \qquad \mathcal{G}_{\text{total}}^{GR} = \mathcal{G}_{GR} \ltimes \tilde{\mathcal{G}}_{HT} \, ,$$

in the sense that neither is a subgroup of the other. This conclusion is often overlooked in the literature, which mostly puts emphasis on the symmetry breaking of the ordinary gauge group down to its subgroup.

Let us state here, without proof, that the action (52) represents an example of a non-topological action, for which one can also demonstrate a property analogous to (40), that diffeomorphisms are not a subgroup of its ordinary gauge group, but are a subgroup of the total gauge group. Simply put, given that the simplicity constraint term in (52) breaks the ordinary gauge symmetry group $\mathcal{G}_{2BF}$ into its subgroup $\mathcal{G}_{GR}$ (see (53)), one can expect that diffeomorphisms are not a subgroup of $\mathcal{G}_{GR}$, since they are not a subgroup of the larger group $\mathcal{G}_{2BF}$ of the topological action (46). Nevertheless, since the action (52) is written in a manifestly covariant form, diffeomorphisms are certainly a symmetry of the action and, thus, must be a subgroup of the total gauge group $\mathcal{G}_{\text{total}}^{GR} = \mathcal{G}_{GR} \ltimes \tilde{\mathcal{G}}_{HT}$, in line with the statement analogous to (40). We leave the details of the proof as an exercise for the reader. The point of this analysis was to demonstrate that the interplay (40) between diffeomorphisms and the HT symmetry is a generic property of a large class of actions, including the physically relevant ones, and not limited to examples of topological theories such as the Chern–Simons or $nBF$ models.

As the last comment, let us remark that, in fact, almost all conclusions discussed for the cases of the Chern–Simons, $3BF$, and $2BF$ theories are not really specific to these concrete cases. One can easily generalize our analysis to any other theory, and the conclusions should remain unchanged, except maybe in some corner cases.

## 5. Conclusions

Let us review the results. In Section 2, we gave a short overview of HT gauge symmetry and discussed its most-important general properties. First, the HT group is a normal subgroup of the total group of gauge symmetries of any given action. Second, HT transformations exhaust all "trivial" (i.e., vanishing on-shell) symmetries, in the sense that there are no trivial symmetries that are not of the HT type. Finally, adding additional terms into the action substantially changes the HT group, often enlarging it. This may be considered a counterintuitive result, since usually adding additional terms in the action serves the purpose of fixing the gauge and, thus, is meant to reduce the gauge symmetry, rather than to enlarge it.

After these general results, in Section 3, we discussed the HT symmetry of the Chern–Simons action, which is a convenient toy example that neatly displays the general features from Section 2. Special attention was given to the issue of diffeomorphisms, and it was shown that, while they are not a subgroup of the ordinary gauge group of the Chern–Simons action, they nevertheless do represent a proper subgroup of the total gauge symmetry, and the HT subgroup plays a nontrivial role in demonstrating this.

Section 4 was devoted to the study of HT symmetry in the $2BF$ and $3BF$ theories, which are relevant for the constructions of realistic quantum gravity models within the generalized spinfoam approach and higher gauge theory. After a brief review and introduction to the notion of three-groups and the $3BF$ theory, appropriate HT transformations were explicitly constructed, complementing the ordinary group of gauge symmetries of the $3BF$ action based on a given three-group. This gave us the total gauge symmetry group for this class

of theories. We again discussed the issue of diffeomorphisms and demonstrated again that they are a subgroup of the total gauge group, without being a subgroup of the ordinary gauge group, just like in the case of the Chern–Simons theory. Finally, we introduced a completely concrete example of the $2BF$ theory based on the Poincaré two-group, which becomes classically equivalent to Einstein's general relativity when one introduces the additional term into the action, called the simplicity constraint. As argued in general in Section 2, the presence of this constraint breaks the ordinary gauge group down into its subgroup, while simultaneously enlarging the HT group, since it introduces an additional Lagrange multiplier field into the action. This represents an explicit example of the general statement from Section 2 that the total gauge symmetry group changes nontrivially, as opposed to simply breaking down to its subgroup.

It should be noted that the analysis and results discussed here do not cover everything that can be said about HT symmetry. Among the topics not covered, one can mention the question of an explicit form of finite HT transformations, as opposed to infinitesimal ones. Can one write down finite HT transformations in closed form, either for some conveniently chosen action or maybe even in general? A related topic is the explicit evaluation of the commutator of two HT transformations, or equivalently, the structure constants of the HT Lie algebra, or in yet other words, the multiplication rule in the group $\mathcal{G}_{HT}$. Is the group Abelian or not and for which choices of the action? Finally, one would also like to know the topological properties of the group $\mathcal{G}_{HT}$, i.e., its global structure. All these are potentially interesting topics for future research.

As a particularly interesting topic for future research, we should mention the nontrivial change of the HT symmetry group when additional terms are being added to the action. In Section 4.4, we briefly demonstrated that HT symmetry does change in a nontrivial way, on the example action (52). Nevertheless, the precise properties and the physical interpretation of this change are yet to be studied in full and for a general choice of the action. This topic is the subject of ongoing research.

Finally, we would like to reiterate the differences in two possible approaches to the notion of "the gauge symmetry" of a theory. The overwhelmingly common approach throughout the literature is to factor out the HT group and work only with the ordinary, nontrivial gauge group as the relevant symmetry. Admittedly, this approach does feature a certain level of appeal due to its simplicity and economy, since it does not have to deal with HT symmetry at all. Nevertheless, there are important situations where this is not enough, and one really needs to take into account the *total* gauge symmetry group, which includes HT transformations. As a rule, these situations always involve the gauge symmetry off-shell, either for the purpose of quantization or otherwise. A typical example is the Batalin–Vilkovisky formalism, where one needs to explicitly keep track of HT transformations throughout the whole analysis. Another situation, which was discussed here in more detail, is the question of diffeomorphism symmetry, where HT transformations are required in order to prove that diffeomorphisms are a symmetry of the theory even off-shell. This is especially relevant for building quantum gravity models. Finally, the third scenario would be the discussion of the Coleman–Mandula theorem. One of the main assumptions of the theorem is that the Poincaré group is a subgroup of the full symmetry group of the theory. Given this assumption, and a number of other assumptions, the theorem implies that the full symmetry group must be a direct product of the Poincaré subgroup and the internal symmetry subgroup. In certain cases of theories (such as the $3BF$ action), the full symmetry group is not explicitly expressed as such a direct product, and moreover, it is not obvious that the Poincaré group is a subgroup of the full symmetry group to begin with. Therefore, in order to verify whether the above assumption of the theorem is satisfied, one needs to inspect if the Poincaré group is or is not a subgroup of the full symmetry group. At this point, one may run into a scenario similar to diffeomorphisms: the Poincaré group may fail to be a subgroup of the ordinary gauge group, but still be a subgroup of the total gauge group, once the HT symmetry is taken into account. In this sense, HT symmetry

may become relevant for the proper analysis and application of the Coleman–Mandula theorem in certain contexts. This topic is the subject of ongoing research [34].

All of the above arguments suggest that it may be prudent to abandon the common approach of factoring out the HT group and instead adopt the description of the symmetry with the total gauge group, which includes HT transformations on equal footing as the ordinary gauge transformations. In the long run, this may be a conceptually cleaner approach. However, either way, we believe that HT symmetry is relevant for the overall symmetry structure of a theory and that better understanding of its properties can add value to and benefit research.

**Author Contributions:** Conceptualization, M.V.; investigation, M.Đ., T.R., P.S. and M.V.; writing—original draft preparation, M.Đ., T.R., P.S. and M.V.; writing—review and editing, M.Đ., T.R., P.S. and M.V. All authors have read and agreed to the published version of the manuscript.

**Funding:** All authors were supported by the Ministry of Science, Technological development and Innovations of the Republic of Serbia. In addition, T.R. and M.V. were supported by the Science Fund of the Republic of Serbia, Grant 7745968, "Quantum Gravity from Higher Gauge Theory 2021"—QGHG-2021. The contents of this publication are the sole responsibility of the authors and can in no way be taken to reflect the views of the Science Fund of the Republic of Serbia.

**Data Availability Statement:** No new data were created nor analyzed in this study. Data sharing is not applicable to this article.

**Acknowledgments:** The authors would like to thank Igor Prlina for discussions and suggestions when writing this manuscript.

**Conflicts of Interest:** The authors declare no conflict of interest.

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
