# Peer review of "Henneaux–Teitelboim Gauge Symmetry and Its Applications to Higher Gauge Theories"

_universe, doi:10.3390/universe9060281_

Round 1

Reviewer 1 Report

In this manuscript the authors discuss the so-called Henneaux-Teitelboim (HT) transformations and their main properties. They also explicitly work through some key examples in order to highlight the potential importance of the HT symmetry and how it can play a crucial role in a variety of contexts. These examples comprise the Chern-Simons theory and a class of topological theories called nBF models. The authors point out that a description which incorporates HT transformations on equal footing as ordinary gauge transformations may be conceptually cleaner.

This is a well written manuscript. Even though the setting is not completely original, this subject is interesting in its own right. I am inclined to recommend it for publication. In any case I would like to draw the authors' attention to some brief comments listed below. 

(1) Despite some efforts by the authors to try to emphasize the conceptual importance of HT transformations, it is still unclear how useful this symmetry can be when discussing, for instance, physical observables. One may argue that in the end one is really interested in on-shell physics; in some theories (for example, string theory in the flat space), only on-shell amplitudes for external particles such as gravitons may be evaluated. In summary, the manuscript lacks a more detailed discussion on how HT transformations may play a relevant role in more general physically relevant situations. This should be of interest to a broader readership.

(2) Usually we deem a given symmetry as a "gauge symmetry" after we establish a procedure of gauging a particular global transformation. So it is unclear that associating HT symmetry with "gauge symmetry" would be the most appropriate terminology here.

(3) As pointed out by the authors, the introduction of additional terms to the action modifies the total gauge symmetry in a non-trivial way. It would be interesting to have a brief discussion about the physical interpretation of this result.

(4) One interesting feature of the examples discussed by the authors is that diffeomorphisms can only be considered as a subgroup of the total gauge group (which includes HT transformations). But it is unclear whether this is an artifact of topological gauge theories. This could be discussed further by the authors.

(5) In the Conclusion section, the discussion on the Coleman-Mandula theorem needs some reassessment — essentially it is not clear that HT symmetry is relevant for the theorem. Indeed, one can think of at least two reasons. One is the statement of the theorem itself: the symmetry group of a (non-conformal, non-supersymmetric) ordinary quantum field theory in four spacetime dimensions is necessarily a direct product of the Poincaré group and an internal symmetry group. On the other hand, one of the basic assumptions of the theorem is that external particles are subject to on-shell constraints, and, as remarked by the authors, HT transformations enjoy on-shell triviality.

Author Response

Responses to Reviewer 1:

> In this manuscript the authors discuss the so-called Henneaux-Teitelboim (HT)
> transformations and their main properties. They also explicitly work through
> some key examples in order to highlight the potential importance of the HT
> symmetry and how it can play a crucial role in a variety of contexts. These
> examples comprise the Chern-Simons theory and a class of topological theories
> called nBF models. The authors point out that a description which incorporates
> HT transformations on equal footing as ordinary gauge transformations may be
> conceptually cleaner.
>  
> This is a well written manuscript. Even though the setting is not completely
> original, this subject is interesting in its own right. I am inclined to
> recommend it for publication. In any case I would like to draw the authors'
> attention to some brief comments listed below.  

We wish to thank the Reviewer for the positive opinion of our manuscript. Regarding the Reviewer's comments, below we answer each one in turn.

> (1) Despite some efforts by the authors to try to emphasize the conceptual
> importance of HT transformations, it is still unclear how useful this symmetry
> can be when discussing, for instance, physical observables. One may argue that
> in the end one is really interested in on-shell physics; in some theories (for
> example, string theory in the flat space), only on-shell amplitudes for
> external particles such as gravitons may be evaluated. In summary, the
> manuscript lacks a more detailed discussion on how HT transformations may play
> a relevant role in more general physically relevant situations. This should be
> of interest to a broader readership.

We fully agree with the Reviewer that HT symmetry is not really relevant for on-shell physics, such as when discussing scattering processes and similar. This is in fact the reason why most authors call this symmetry "trivial". However, we also believe that on-shell physics is mostly relevant when applying an existing theory to extract predictions, while off-shell properties play a more prominent role when one is trying to construct a new theory. A typical example is the quantization of a classical theory --- different off-shell versions of the same classical theory typically lead to different quantum theories (this can be easily seen from the path integral quantization prescription, which depends not only on the classical equations of motion, but on the whole classical action). Related to this is also the Batalin-Vilkovisky formalism, in which HT transformations play a very substantial role.

In other words, HT symmetry is not relevant for making predictions, but it does have methodological value in theory construction. We agree with the Reviewer that this was not emphasized enough in the old version of the manuscript. In the new version, we have tried to elaborate a bit more what is the actual benefit of studying HT symmetry, to make it more clear for the readers.

> (2) Usually we deem a given symmetry as a "gauge symmetry" after we establish
> a procedure of gauging a particular global transformation. So it is unclear
> that associating HT symmetry with "gauge symmetry" would be the most
> appropriate terminology here.

We agree with the Reviewer that the terminology used in the manuscript may not be the most adequate. However, throughout the literature, there is no standardized terminology which clearly distinguishes between local symmetries which have been obtained by localization (gauging) some global symmetry, versus local symmetries which have not been obtained by any such localization. Indeed, for the case of HT symmetry, it is not even clear whether or not there exists a corresponding global symmetry group, such that its localization yields the discussed HT transformations.

In the new version of the manuscript, we have added a comment warning the reader about our use of terminology. However, we are at a loss for any alternative choice of terminology that would be more suitable, so we opted to keep the same one, describing the set of HT transformations as a gauge symmetry. The comment for the reader should be enough to eliminate any confusion about the meaning.

> (3) As pointed out by the authors, the introduction of additional terms to the
> action modifies the total gauge symmetry in a non-trivial way. It would be
> interesting to have a brief discussion about the physical interpretation of
> this result.

This is indeed a very interesting question, and in fact one of our current research topics is related to the study of adding additional terms to the nBF class of topological actions. These terms deform a topological theory into a non-topological one, introducing local propagating degrees of freedom. Within the scope of this research, we also study how HT symmetry is being changed by the deformation of the action, and what are the physical consequences of this change.

In other words, this is a topic of our ongoing research, and is somewhat out of the scope of the current manuscript. We have added a brief comment about this in the new version of the manuscript (as one of the future lines of investigation), but we refrain from discussing any specific details regarding the physical interpretation of the change of HT symmetry, leaving this topic for future work.

> (4) One interesting feature of the examples discussed by the authors is that
> diffeomorphisms can only be considered as a subgroup of the total gauge group
> (which includes HT transformations). But it is unclear whether this is an
> artifact of topological gauge theories. This could be discussed further by the
> authors.

This is also a very interesting question, and again one which we will address in our future paper (see previous comment). But here we can give a short answer to Reviewer's question --- the relationship between diffeomorphisms and HT transformations is not merely an artifact of topological theories. There are explicit examples of non-topological theories where the same effect happens, namely diffeomorphisms are not a subgroup of ordinary gauge group but are a subgroup of the total gauge group (which includes HT transformations). One such example is the action (52), which is classically equivalent to general relativity, i.e., is not topological.

That said, the systematic discussion of such non-topological theories is out of the scope of the current manuscript, and they will be explicitly studied in our future paper. In the new version of the current manuscript, we have instead added just a short analysis for the case of the action (52), without proof, as an example for the readers.

> (5) In the Conclusion section, the discussion on the Coleman-Mandula theorem
> needs some reassessment - essentially it is not clear that HT symmetry is
> relevant for the theorem. Indeed, one can think of at least two reasons. One
> is the statement of the theorem itself: the symmetry group of a (non-conformal,
> non-supersymmetric) ordinary quantum field theory in four spacetime dimensions
> is necessarily a direct product of the Poincaré group and an internal symmetry
> group. On the other hand, one of the basic assumptions of the theorem is that
> external particles are subject to on-shell constraints, and, as remarked by
> the authors, HT transformations enjoy on-shell triviality.

We agree with both statements given by the Reviewer. Nevertheless, the relevance of HT symmetry for the Coleman-Mandula (CM) theorem lies elsewhere. Namely, in the actual formulation of the theorem, one of the assumptions is that the Poincare group is a subgroup of the full symmetry group of the theory (the latter being defined as a set of all operators which commute with the S-matrix). Given this assumption (and a number of others), the theorem then infers that this full symmetry group can be written as a direct product of the Poincare and internal symmetry subgroups.

The importance of HT transformations lies in the above assumption, which can be seen as follows. Formally, given a theory with some symmetry group, before applying the CM theorem, one first needs to verify whether the above assumption is satisfied, i.e., whether the Poincare group is indeed a subgroup of the full symmetry group. It may happen (and in certain examples it does happen) that Poincare group displays the same behavior as diffeomorphisms --- it is *not* a subgroup of ordinary gauge group, but *is* a subgroup of the total gauge group, with HT transformations included. In this sense, HT symmetry can become relevant when testing the assumptions that enter the CM theorem. In fact, in our separate research, we have managed to run precisely into this issue when we attempted to apply the CM theorem to nBF class of actions (and as noted in the manuscript, we are preparing a separate paper that focuses precisely on this topic).

Of course, we agree with the Reviewer that the discussion regarding the CM theorem is not precise enough in the old version of the manuscript. To that end, in the new version we have expanded that part in order to explain the relevance of HT transformations for the CM theorem more clearly.

Finally, we would like to thank the Reviewer for all comments and suggestions, since they helped improve the overall presentation of the material in the manuscript.

===================
List of changes to the manuscript:

* References [2-6] added (in response to the comment by Reviewer 2).

* Third paragraph added to the Introduction section (in response to the comment (1) by Reviewer 1).

* Sixth paragraph added to the Introduction section (in response to the comment (2) by Reviewer 1).

* Fifth paragraph added to the Conclusions section (in response to the comment (3) by Reviewer 1).

* Penultimate paragraph added to Section 4 (in response to the comment (4) by Reviewer 1).

* Penultimate paragraph of the Conclusions section modified and expanded (in response to the comment (5) by Reviewer 1).

Reviewer 2 Report

The paper under consideration is devoted to review of special type of gauge transformations, which are called by authors as Henneaux-Teitelboim transformations. The definition of the these transformation is given and the their properties are considered. The various applications of such transformations in gauge field theory and in quantum gravity, in particular, are also discussed. The paper is written in clear form with all appropriare details. It is a useful review and I recommend it for publications. It is a useful review and I recommend it for publications. There is the only commnet, the transformations under review were appeared in the papers by Batalin and Vilkovisky in context of theories with open gauge algebra and I think it is worth noting  this point.

Author Response

Response to Reviewer 2:

> The paper under consideration is devoted to review of special type of gauge
> transformations, which are called by authors as Henneaux-Teitelboim
> transformations. The definition of the these transformation is given and the
> their properties are considered. The various applications of such
> transformations in gauge field theory and in quantum gravity, in particular,
> are also discussed. The paper is written in clear form with all appropriare
> details. It is a useful review and I recommend it for publications. It is a
> useful review and I recommend it for publications. There is the only commnet,
> the transformations under review were appeared in the papers by Batalin and
> Vilkovisky in context of theories with open gauge algebra and I think it is
> worth noting  this point.

We wish to thank the Reviewer for the positive opinion of our manuscript. In the new version of the manuscript, we have added a comment and included the references to the original papers by Batalin and Vilkovisky.

===================
List of changes to the manuscript:

* References [2-6] added (in response to the comment by Reviewer 2).

* Third paragraph added to the Introduction section (in response to the comment (1) by Reviewer 1).

* Sixth paragraph added to the Introduction section (in response to the comment (2) by Reviewer 1).

* Fifth paragraph added to the Conclusions section (in response to the comment (3) by Reviewer 1).

* Penultimate paragraph added to Section 4 (in response to the comment (4) by Reviewer 1).

* Penultimate paragraph of the Conclusions section modified and expanded (in response to the comment (5) by Reviewer 1).

Round 2

Reviewer 1 Report

The authors have addressed in detail all points raised previously. More elaborate discussions were given. This is an improved version of the work in comparison with the previous one. I am satisfied with the science and the presentation. Therefore I recommend the present manuscript for publication.